# Auxiliary-variable Exact Hamiltonian Monte Carlo Samplers for Binary Distributions

**Ari Pakman and Liam Paninski**
Department of Statistics
Center for Theoretical Neuroscience
Grossman Center for the Statistics of Mind
Columbia University
New York, NY, 10027

## Abstract

We present a new approach to sample from generic binary distributions, based on an exact Hamiltonian Monte Carlo algorithm applied to a piecewise continuous augmentation of the binary distribution of interest. An extension of this idea to distributions over mixtures of binary and possibly-truncated Gaussian or exponential variables allows us to sample from posteriors of linear and probit regression models with spike-and-slab priors and truncated parameters. We illustrate the advantages of these algorithms in several examples in which they outperform the Metropolis or Gibbs samplers.

## 1 Introduction

Mapping a problem involving discrete variables into continuous variables often results in a more tractable formulation. For the case of probabilistic inference, in this paper we present a new approach to sample from distributions over binary variables, based on mapping the original discrete distribution into a continuous one with a piecewise quadratic log-likelihood, from which we can sample efficiently using exact Hamiltonian Monte Carlo (HMC).

The HMC method is a Markov Chain Monte Carlo algorithm that usually has better performance over Metropolis or Gibbs samplers, because it manages to propose transitions in the Markov chain which lie far apart in the sampling space, while maintaining a reasonable acceptance rate for these proposals. But the implementations of HMC algorithms generally involve the non-trivial tuning of numerical integration parameters to obtain such a reasonable acceptance rate (see [1] for a review). The algorithms we present in this work are special because the Hamiltonian equations of motion can be integrated exactly, so there is no need for tuning a step-size parameter and the Markov chain always accepts the proposed moves. Similar ideas have been used recently to sample from truncated Gaussian multivariate distributions [2], allowing much faster sampling than other methods. It should be emphasized that despite the apparent complexity of deriving the new algorithms, their implementation is very simple.

Since the method we present transforms a binary sampling problem into a continuous one, it is natural to extend it to distributions defined over mixtures of binary and Gaussian or exponential variables, transforming them into purely continuous distributions. Such a mixed binary-continuous problem arises in Bayesian model selection with a spike-and-slab prior and we illustrate our technique by focusing on this case. In particular, we show how to sample from the posterior of linear and probit regression models with spike-and-slab priors, while also imposing truncations in the parameter space (e.g., positivity).

The method we use to map binary to continuous variables consists in simply identifying a binary variable with the sign of a continuous one. An alternative relaxation of binary to continuous vari-

ables, known in statistical physics as the "Gaussian integral trick" [3], has been used recently to apply HMC methods to binary distributions [4], but the details of that method are different than ours. In particular, the HMC in that work is not 'exact' in the sense used above and the algorithm only works for Markov random fields with Gaussian potentials.

## 2 Binary distributions

We are interested in sampling from a probability distribution $p(\mathbf{s})$ defined over $d$-dimensional binary vectors $\mathbf{s} \in \{-1, +1\}^d$, and given in terms of a function $f(\mathbf{s})$ as

$$p(\mathbf{s}) = \frac{1}{Z} f(\mathbf{s}) \,. \tag{1}$$

Here $Z$ is a normalization factor, whose value will not be needed. Let us augment the distribution $p(\mathbf{s})$ with continuous variables $\mathbf{y} \in \mathbb{R}^d$ as

$$p(\mathbf{s}, \mathbf{y}) = p(\mathbf{s})p(\mathbf{y}|\mathbf{s}) \tag{2}$$

where $p(\mathbf{y}|\mathbf{s})$ is non-zero only in the orthant defined by

$$s_i = sign(y_i) \qquad i = 1, \ldots, d. \tag{3}$$

The essence of the proposed method is that we can sample from $p(\mathbf{s})$ by sampling $\mathbf{y}$ from

$$\begin{aligned} p(\mathbf{y}) &= \sum_{\mathbf{s}'} p(\mathbf{s}')p(\mathbf{y}|\mathbf{s}') \,, &&(4) \\ &= p(\mathbf{s})p(\mathbf{y}|\mathbf{s}) \,, &&(5) \end{aligned}$$

and reading out the values of $\mathbf{s}$ from (3). In the second line we have made explicit that for each $\mathbf{y}$, only one term in the sum in (4) is non-zero, so that $p(\mathbf{y})$ is piecewise defined in each orthant.

In order to sample from $p(\mathbf{y})$ using the exact HMC method of [2], we require $\log p(\mathbf{y}|\mathbf{s})$ to be a quadratic function of $\mathbf{y}$ on its support. The idea is to define a potential energy function

$$U(\mathbf{y}) = -\log p(\mathbf{y}|\mathbf{s}) - \log f(\mathbf{s}) \,, \tag{6}$$

introduce momentum variables $q_i$, and consider the piecewise continuous Hamiltonian

$$H(\mathbf{y}, \mathbf{q}) \quad = U(\mathbf{y}) + \frac{\mathbf{q} \cdot \mathbf{q}}{2} \,, \tag{7}$$

whose value is identified with the energy of a particle moving in a $d$-dimensional space. Suppose the particle has initial coordinates $\mathbf{y}(0)$. In each iteration of the sampling algorithm, we sample initial values $\mathbf{q}(0)$ for the momenta from a standard Gaussian distribution and let the particle move during a time $T$ according to the equations of motion

$$\dot{\mathbf{y}}(t) = \frac{\partial H}{\partial \mathbf{q}(t)} \,, \qquad \dot{\mathbf{q}}(t) = -\frac{\partial H}{\partial \mathbf{y}(t)} \,. \tag{8}$$

The final coordinates, $\mathbf{y}(T)$, belong to a Markov chain with invariant distribution $p(\mathbf{y})$, and are used as the initial coordinates of the next iteration. The detailed balance condition follows directly from the conservation of energy and $(\mathbf{y}, \mathbf{q})$-volume along the trajectory dictated by (8), see [1, 2] for details.

The restriction to quadratic functions of $\mathbf{y}$ in $\log p(\mathbf{y}|\mathbf{s})$ allows us to solve the differential equations (8) exactly in each orthant. As the particle moves, the potential energy $U(\mathbf{y})$ and the kinetic energy $\frac{\mathbf{q} \cdot \mathbf{q}}{2}$ change in tandem, keeping the value of the Hamiltonian (7) constant. But this smooth interchange gets interrupted when any coordinate reaches zero. Suppose this first happens at time $t_j$ for coordinate $y_j$, and assume that $y_j < 0$ for $t < t_j$. Conservation of energy imposes now a jump on the momentum $q_j$ as a result of the discontinuity in $U(\mathbf{y})$. Let us call $q_j(t_j^-)$ and $q_j(t_j^+)$ the value of the momentum $q_j$ just before and after the coordinate hits $y_j = 0$. In order to enforce conservation of energy, we equate the Hamiltonian at both sides of the $y_j = 0$ wall, giving

$$\frac{q_j^2(t_j^+)}{2} = \Delta_j + \frac{q_j^2(t_j^-)}{2} \tag{9}$$

with

$$\Delta_j = U(y_j = 0, s_j = -1) - U(y_j = 0, s_j = +1) \tag{10}$$

If eq. (9) gives a positive value for $q_j^2(t_j^+)$, the coordinate $y_j$ crosses the boundary and continues its trajectory in the new orthant. On the other hand, if eq.(9) gives a negative value for $q_j^2(t_j^+)$, the particle is reflected from the $y_j = 0$ wall and continues its trajectory with $q_j(t_j^+) = -q_j(t_j^-)$. When $\Delta_j < 0$, the situation can be understood as the limit of a scenario in which the particle faces an upward hill in the potential energy, causing it to diminish its velocity until it either reaches the top of the hill with a lower velocity or stops and then reverses. In the limit in which the hill has finite height but infinite slope, the velocity change occurs discontinuously at one instant. Note that we used in (9) that the momenta $q_{i \neq j}$ are continuous, since this sudden infinite slope hill is only seen by the $y_j$ coordinate.

Regardless of whether the particle bounces or crosses the $y_j = 0$ wall, the other coordinates move unperturbed until the next boundary hit, where a similar crossing or reflection occurs, and so on, until the final position $\mathbf{y}(T)$.

The framework we presented above is very general and in order to implement a particular sampler we need to select the distributions $p(\mathbf{y}|\mathbf{s})$. Below we consider in some detail two particularly simple choices that illustrate the diversity of options here.

## 2.1 Gaussian augmentation

Let us consider first for $p(\mathbf{y}|\mathbf{s})$ the truncated Gaussians

$$p(\mathbf{y}|\mathbf{s}) = \begin{cases} (2/\pi)^{d/2} \, e^{-\frac{\mathbf{y} \cdot \mathbf{y}}{2}} & \text{for } sign(y_i) = s_i, \qquad i = 1, \dots, d \\ 0 & \text{otherwise} , \end{cases} \tag{11}$$

The equations of motion (8) lead to $\ddot{\mathbf{y}}(t) = -\mathbf{y}(t)$, $\ddot{\mathbf{q}}(t) = -\mathbf{q}(t)$, and have a solution

$$\begin{align} y_i(t) &= y_i(0)\cos(t) + q_i(0)\sin(t) , \tag{12} \\ &= u_i \sin(\phi_i + t) , \tag{13} \\ q_i(t) &= -y_i(0)\sin(t) + q_i(0)\cos(t) , \tag{14} \\ &= u_i \cos(\phi_i + t) . \tag{15} \end{align}$$

This setting is similar to the case studied in [2] and from $\phi_i = \tan^{-1}(y_i(0)/q_i(0))$ the boundary hit times $t_i$ are easily obtained. When a boundary is reached, say $y_j = 0$, the coordinate $y_j$ changes its trajectory for $t > t_j$ as

$$y_j(t) = q_j(t_j^+)\sin(t - t_j) , \tag{16}$$

with the value of $q_j(t_j^+)$ obtained as described above.

Choosing an appropriate value for the travel time $T$ is crucial when using HMC algorithms [5]. As is clear from (13), if we let the particle travel during a time $T > \pi$, each coordinate reaches zero at least once, and the hitting times can be ordered as

$$0 < t_{j_1} \leq t_{j_2} \leq \cdots \leq t_{j_d} < \pi . \tag{17}$$

Moreover, regardless of whether a coordinate crosses zero or gets reflected, it follows from (16) that the successive hits occur at

$$t_i + n\pi, \quad n = 1, 2, \dots \tag{18}$$

and therefore the hitting times only need to be computed once for each coordinate in every iteration. If we let the particle move during a time $T = n\pi$, each coordinate reaches zero $n$ times, in the cyclical order (17), with a computational cost of $O(nd)$ from wall hits. But choosing *precisely* $T = n\pi$ is not recommended for the following reason. As we just showed, between $y_j(0)$ and $y_j(\pi)$ the coordinate touches the boundary $y_j = 0$ once, and if $y_j$ gets reflected off the boundary, it is easy to check that we have $y_j(\pi) = y_j(0)$. If we take $T = n\pi$ and the particle gets reflected all the $n$ times it hits the boundary, we get $y_j(T) = y_j(0)$ and the coordinate $y_j$ does not move at all. To avoid these singular situations, a good choice is $T = (n + \frac{1}{2})\pi$, which generalizes the recommended

travel time $T = \pi/2$ for truncated Gaussians in [2]. The value of $n$ should be chosen for each distribution, but we expect optimal values for $n$ to grow with $d$.

With $T = (n + \frac{1}{2})\pi$, the total cost of each sample is $O((n + 1/2)d)$ on average from wall hits, plus $O(d)$ from the sampling of $\mathbf{q}(0)$ and from the $d$ inverse trigonometric functions to obtain the hit times $t_i$. But in complex distributions, the computational cost is dominated by the the evaluation of $\Delta_i$ in (10) at each wall hit.

Interestingly, the rate at which wall $y_i = 0$ is crossed coincides with the acceptance rate in a Metropolis algorithm that samples uniformly a value for $i$ and makes a proposal of flipping the binary variable $s_i$. See the Appendix for details. Of course, this does not mean that the HMC algorithm is the same as Metropolis, because in HMC the order in which the walls are hit is fixed given the initial velocity, and the values of $q_i^2$ at successive hits of $y_i = 0$ within the same iteration are not independent.

## 2.2 Exponential and other augmentations

Another distribution that allows one an exact solution of the equations of motion (8) is

$$p(\mathbf{y}|\mathbf{s}) = \begin{cases} e^{-\sum_{i=1}^{d} |y_i|} & \text{for } sign(y_i) = s_i, \qquad i = 1, \dots, d \\ 0 & \text{otherwise}, \end{cases} \tag{19}$$

which leads to the equations of motion $\ddot{y}_i(t) = -s_i$, with solutions of the form

$$y_i(t) \quad = \quad y_i(0) + q_i(0)t - \frac{s_i t^2}{2} . \tag{20}$$

In this case, the initial hit time for every coordinate is the solution of the quadratic equation $y_i(t) = 0$. But, unlike the case of the Gaussian augmentation, the order of successive hits is not fixed. Indeed, if coordinate $y_j$ hits zero at time $t_j$, it continues its trajectory as

$$y_j(t > t_j) = q(t_j^+)(t - t_j) - \frac{s_j}{2}(t - t_j)^2 , \tag{21}$$

so the next wall hit $y_j = 0$ will occur at a time $t_j'$ given by

$$(t_j' - t_j) = 2|q_j(t_j^+)| , \tag{22}$$

where we used $s_j = sign(q_j(t_j^+))$. So we see that the time between successive hits of the same coordinate depends only on its momentum after the last hit. Moreover, since the value of $|q_j(t^+)|$ is smaller than $|q_j(t^-)|$ if the coordinate crosses to an orthant of lower probability, equation (22) implies that the particle moves away faster from areas of lower probability. This is unlike the Gaussian augmentation, where a coordinate 'waits in line' until all the other coordinates touch their wall before hitting its wall again.

The two augmentations we considered above have only scratched the surface of interesting possibilities. One could also define $f(\mathbf{y}|\mathbf{s})$ as a uniform distribution in a box such that the computation of the times for wall hits would becomes purely linear and we get a classical 'billiards' dynamics. More generally, one could consider different augmentations in different orthants and potentially tailor the algorithm to mix faster in complex and multimodal distributions.

## 3 Spike-and-slab regression with truncated parameters

The subject of Bayesian sparse regression has seen a lot of work during the last decade. Along with priors such as the Bayesian Lasso [6] and the Horsehoe [7], the classic spike-and-slab prior [8, 9] still remains very competitive, as shown by its superior performance in the recent works [10, 11, 12]. But despite its successes, posterior inference remains a computational challenge for the spike-and-slab prior. In this section we will show how the HMC binary sampler can be extended to sample from the posterior of these models. The latter is a distribution over a set of binary and continuous variables, with the binary variables determining whether each coefficient should be included in the model or not. The idea is to map these indicator binary variables into continuous variables as we did above, obtaining a distribution from which we can sample again using exact HMC methods. Below we consider a regression model with Gaussian noise but the extension to exponential noise (or other scale-mixtures of Gaussians) is immediate.

### 3.1 Linear regression

Consider a regression problem with a log-likelihood that depends quadratically on its coefficients, such as

$$\log p(D|\mathbf{w}) = -\frac{1}{2}\mathbf{w}'\mathbf{M}\mathbf{w} + \mathbf{r}\cdot\mathbf{w} + const. \tag{23}$$

where $D$ represents the observed data. In a linear regression model $\mathbf{z} = X\mathbf{w}+\varepsilon$, with $\varepsilon \sim \mathcal{N}(0,\sigma^2)$, we have $\mathbf{M} = X'X/\sigma^2$ and $\mathbf{r} = \mathbf{z}'X/\sigma^2$. We are interested in a spike-and-slab prior for the coefficients $\mathbf{w}$ of the form

$$p(\mathbf{w},\mathbf{s}|a,\tau^2) = \prod_{i=1}^{d} p(w_i|s_i,\tau^2)p(s_i|a)\,. \tag{24}$$

Each binary variable $s_i = \pm 1$ has a Bernoulli prior $p(s_i|a) = a^{\frac{(1+s_i)}{2}}(1-a)^{\frac{(1-s_i)}{2}}$ and determines whether the coefficient $w_i$ is included in the model. The prior for $w_i$, conditioned on $s_i$, is

$$p(w_i|s_i,\tau^2) = \begin{cases} \frac{1}{\sqrt{2\pi\tau^2}}e^{-\frac{w_i^2}{2\tau^2}} & \text{for } s_i = +1, \\[2mm] \delta(w_i) & \text{for } s_i = -1 \end{cases} \tag{25}$$

We are interested in sampling from the posterior, given by

$$p(\mathbf{w},\mathbf{s}|D,a,\tau^2) \quad \propto \quad p(D|\mathbf{w})p(\mathbf{w},\mathbf{s}|a,\tau^2) \tag{26}$$

$$\propto \quad \frac{e^{-\frac{1}{2}\mathbf{w}'\mathbf{M}\mathbf{w}+\mathbf{r}\cdot\mathbf{w}}e^{-\frac{1}{2}\mathbf{w}'_+\mathbf{w}_+\tau^{-2}}}{(2\pi\tau^2)^{|\mathbf{s}^+|/2}}\delta(\mathbf{w}_-)a^{|\mathbf{s}^+|}(1-a)^{|\mathbf{s}^-|} \tag{27}$$

$$\propto \quad \frac{e^{-\frac{1}{2}\mathbf{w}'_+\left(\mathbf{M}_++\tau^{-2}\right)\mathbf{w}_++\mathbf{r}_+\cdot\mathbf{w}_+}}{(2\pi\tau^2)^{|\mathbf{s}^+|/2}}\delta(\mathbf{w}_-)a^{|\mathbf{s}^+|}(1-a)^{|\mathbf{s}^-|} \tag{28}$$

where $\mathbf{s}^+$ are the variables with $s_i = +1$ and $\mathbf{s}^-$ those with $s_i = -1$. The notation $\mathbf{r}_+$, $\mathbf{M}_+$ and $\mathbf{w}_+$ indicates a restriction to the $\mathbf{s}^+$ subspace and $\mathbf{w}_-$ indicates a restriction to the $\mathbf{s}^-$ space. In the passage from (27) to (28) we see that the spike-and-slab prior shrinks the dimension of the Gaussian likelihood from $d$ to $|\mathbf{s}^+|$. In principle we could integrate out the weights $\mathbf{w}$ and obtain a collapsed distribution for $\mathbf{s}$, but we are interested in cases in which the space of $\mathbf{w}$ is truncated and therefore the integration is not feasible. An example would be when a non-negativity constraint $w_i \geq 0$ is imposed.

In these cases, one possible approach is to sample from (28) with a block Gibbs sampler over the pairs $\{w_i, s_i\}$, as proposed in [10]. Here we will present an alternative method, extending the ideas of the previous section. For this, we consider a new distribution, obtained in two steps. Firstly, we replace the delta functions in (28) by a factor similar to the slab (25)

$$\delta(w_i) \to \frac{1}{\sqrt{2\pi\tau^2}}e^{-\frac{w_i^2}{2\tau^2}} \qquad s_i = -1 \tag{29}$$

The introduction of a non-singular distribution for those $w_i$'s that are excluded from the model in (28) creates a Reversible Jump sampler [13]: the Markov chain can now keep track of all the coefficients, whether they belong or not to the model in a given state of the chain, thus allowing them to join or leave the model along the chain in a *reversible* way.

Secondly, we augment the distribution with $\mathbf{y}$ variables as in (2)-(5) and sum over $\mathbf{s}$. Using the Gaussian augmentation (11), this gives a distribution

$$p(\mathbf{w},\mathbf{y}|D,a,\tau^2) \propto e^{-\frac{1}{2}\mathbf{w}'_+\left(\mathbf{M}_++\tau^{-2}\right)\mathbf{w}_++\mathbf{r}_+\cdot\mathbf{w}_+}e^{-\frac{\mathbf{w}_-\cdot\mathbf{w}_-}{2\tau^2}}e^{-\frac{\mathbf{y}\cdot\mathbf{y}}{2}}a^{|\mathbf{s}^+|}(1-a)^{|\mathbf{s}^-|} \tag{30}$$

where the values of $\mathbf{s}$ in the rhs are obtained from the signs of $\mathbf{y}$. This is a piecewise Gaussian, different in each orthant of $\mathbf{y}$, and possibly truncated in the $\mathbf{w}$ space. Note that the changes in $p(\mathbf{w},\mathbf{y}|D,a,\tau^2)$ across orthants of $\mathbf{y}$ come both from the factors $a^{|\mathbf{s}^+|}(1-a)^{|\mathbf{s}^-|}$ and from the functional dependence on the $\mathbf{w}$ variables. Sampling from (30) gives us samples from the original distribution (28) using a simple rule: each pair $(w_i, y_i)$ becomes $(w_i, s_i = +1)$ if $y_i \geq 0$ and

$(w_i = 0, s_i = -1)$ if $y_i < 0$. This undoes the steps we took to transform (28) into (30): the identification $s_i = sign(y_i)$ takes us from $p(\mathbf{w}, \mathbf{y}|D, a, \tau^2)$ to $p(\mathbf{w}, \mathbf{s}|D, a, \tau^2)$, and setting $w_i = 0$ when $s_i = -1$ undoes the replacement in (29).

Since (30) is a piecewise Gaussian distribution, we can sample from it again using the methods of [2]. As in that work, the possible truncations for $\mathbf{w}$ are given as $g_n(\mathbf{w}) \geq 0$ for $n = 1, \ldots, N$, with $g_n(\mathbf{w})$ any product of linear and quadratic functions of $\mathbf{w}$. The details are a simple extension of the purely binary case and are not very illuminating, so we leave them for the Appendix.

## 3.2 Probit regression

Consider a probit regression model in which binary variables $b_i = \pm 1$ are observed with probability

$$p(b_i|\mathbf{w}, \mathbf{x}_i) = \frac{1}{\sqrt{2\pi}} \int_{z_i b_i \geq 0} dz_i e^{-\frac{1}{2}(z_i + \mathbf{x}_i \mathbf{w})^2} \tag{31}$$

Given a set of $N$ pairs $(b_i, \mathbf{x}_i)$, we are interested in the posterior distribution of the weights $\mathbf{w}$ using the spike-and-slab prior (24). This posterior is the marginal over the $z_i$'s of the distribution

$$p(\mathbf{z}, \mathbf{w}, \mathbf{s}|\mathbf{x}, a, \tau^2) \propto \prod_{i=1}^{N} e^{-\frac{1}{2}(z_i + \mathbf{x}_i \mathbf{w})^2} p(\mathbf{w}, \mathbf{s}|a, \tau^2) \qquad z_i b_i \geq 0\,, \tag{32}$$

and we can use the same approach as above to transform this distribution into a truncated piecewise Gaussian, defined over the $(N + 2d)$-dimensional space of the vector $(\mathbf{z}, \mathbf{w}, \mathbf{y})$. Each $z_i$ is truncated according to the sign of $b_i$ and we can also truncate the $\mathbf{w}$ space if we so desire. We omit the details of the HMC algorithm, since it is very similar to the linear regression case.

# 4 Examples

We present here three examples that illustrate the advantages of the proposed HMC algorithms over Metropolis or Gibbs samplers.

## 4.1 1D Ising model

We consider a 1D periodic Ising model, with $p(\mathbf{s}) \propto e^{-\beta E[\mathbf{s}]}$, where the energy is $E[\mathbf{s}] = -\sum_{i=1}^{d} s_i s_{i+1}$, with $s_{d+1} = s_1$ and $\beta$ is the inverse temperature. Figure 1 shows the first 1000 iterations of both the Gaussian HMC and the Metropolis[1] sampler on a model with $d = 400$ and $\beta = 0.42$, initialized with all spins $s_i = 1$. In HMC we took a travel time $T = 12.5\pi$ and, for the sake of comparable computational costs, for the Metropolis sampler we recorded the value of $\mathbf{s}$ every $d \times 12.5$ flip proposals. The plot shows clearly that the Markov chain mixes much faster with HMC than with Metropolis. A useful variable that summarizes the behavior of the Markov chain is the magnetization $m = \frac{1}{d} \sum_{i=1}^{d} s_i$, whose expected value is $\langle m \rangle = 0$. The oscillations of both samplers around this value illustrate the superiority of the HMC sampler. In the Appendix we present a more detailed comparison of the HMC Gaussian and exponential and the Metropolis samplers, showing that the Gaussian HMC sampler is the most efficient among the three.

## 4.2 2D Ising model

We consider now a 2D Ising model on a square lattice of size $L \times L$ with periodic boundary conditions below the critical temperature. Starting from a completely disordered state, we compare the time it takes for the sampler to reach one of the two low energy states with magnetization $m \simeq \pm 1$. Figure 2 show the results of 20 simulations of such a model with $L = 100$ and inverse temperature $\beta = 0.5$. We used a Gaussian HMC with $T = 2.5\pi$ and a Metropolis sampler recording values of $\mathbf{s}$ every $2.5L^2$ flip proposals. In general we see that the HMC sampler reaches higher likelihood regions faster.

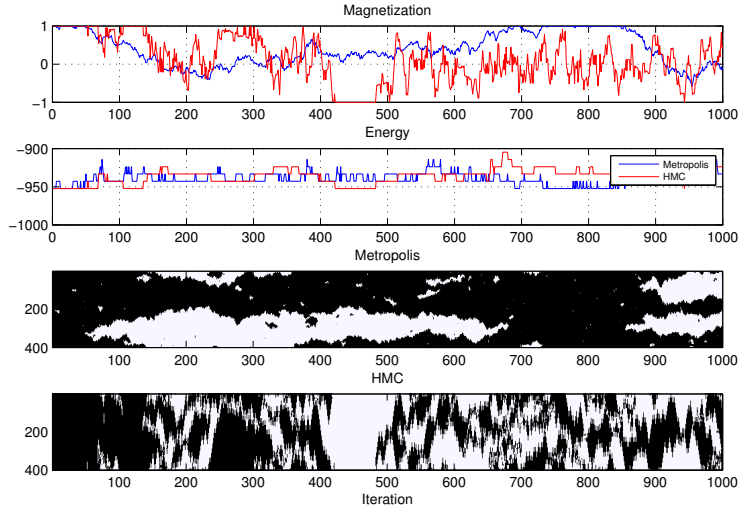

Figure 1: **1D Ising model.** First 1000 iterations of Gaussian HMC and Metropolis samplers on a model with $d = 400$ and $\beta = 0.42$, initialized with all spins $s_i = 1$ (black dots). For HMC the travel time was $T = 12.5\pi$ and in the Metropolis sampler we recorded the state of the Markov chain once every $d \times 12.5$ flip proposals. The lower two panels show the state of $s$ at every iteration for each sampler. The plots show clearly that the HMC model mixes faster than Metropolis in this model.

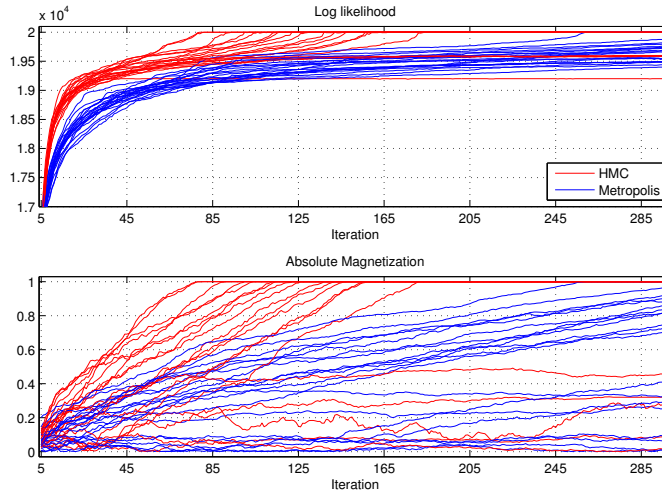

Figure 2: **2D Ising model.** First samples from 20 simulations in a 2D Ising model in a square lattice of side length $L = 100$ with periodic boundary conditions and inverse temperature $\beta = 0.5$. The initial state is totally disordered. We do not show the first 4 samples in order to ease the visualization. For the Gaussian HMC we used $T = 2.5\pi$ and for Metropolis we recorded the state of the chain every $2.5L^2$ flip proposals. The plots illustrate that in general HMC reaches equilibrium faster than Metropolis in this model.

Note that these results of the 1D and 2D Ising models illustrate the advantage of the HMC method in relation to two different time constants relevant for Markov chains [15]. Figure 1 shows that the HMC sampler explores faster the sampled space once the chain has reached its equilibrium distribution, while Figure 2 shows that the HMC sampler is faster in reaching the equilibrium distribution.

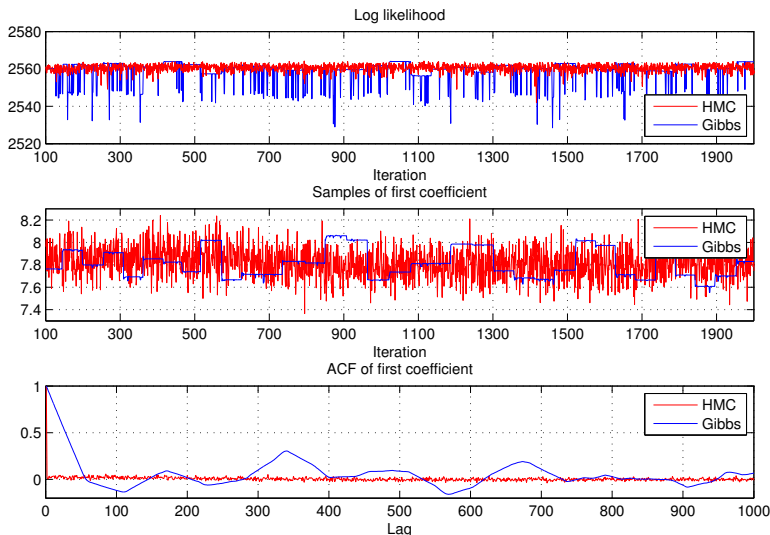

Figure 3: **Spike-and-slab linear regression with constraints.** Comparison of the proposed HMC method with the Gibbs sampler of [10] for the posterior of a linear regression model with spike-and-slab prior, with a positivity constraint on the coefficients. See the text for details of the synthetic data used. Above: log-likelihood as a function of the iteration. Middle: samples of the first coefficient. Below: ACF of the first coefficient. The plots shows clearly that HMC mixes much faster than Gibbs and is more consistent in exploring areas of high probability.

### 4.3 Spike-and-slab linear regression with positive coefficients

We consider a linear regression model $\mathbf{z} = X\mathbf{w} + \varepsilon$ with the following synthetic data. $X$ has $N = 700$ rows, each sampled from a $d = 150$-dimensional Gaussian whose covariance matrix has 3 in the diagonal and 0.3 in the nondiagonal entries. The noise is $\varepsilon \sim \mathcal{N}(0, \sigma^2 = 100)$. The data $\mathbf{z}$ is generated with a coefficients vector $\mathbf{w}$, with 10 non-zero entries with values between 1 and 10. The spike-and-slab hyperparameters are set to $a = 0.1$ and $\tau = 10$. Figure 3 compares the results of the proposed HMC method versus the Gibbs sampler used in [10]. In both cases we impose a positivity constraint on the coefficients. For the HMC sampler we use a travel time $T = \pi/2$. This results in a number of wall hits (both for $\mathbf{w}$ and $\mathbf{y}$ variables) of $\simeq 150$, which makes the computational cost of every HMC and Gibbs sample similar. The advantage of the HMC method is clear, both in exploring regions of higher probability and in the mixing speed of the sampled coefficients. This impressive difference in the efficiency of HMC versus Gibbs is similar to the case of truncated multivariate Gaussians studied in [2].

## 5 Conclusions and outlook

We have presented a novel approach to use exact HMC methods to sample from generic binary distributions and certain distributions over mixed binary and continuous variables,

Even though with the HMC algorithm is better than Metropolis or Gibbs in the examples we presented, this will clearly not be the case in many complex binary distributions for which specialized sampling algorithms have been developed, such as the Wolff or Swendsen-Wang algorithms for 2D Ising models near the critical temperature [14]. But in particularly difficult distributions, these HMC algorithms could be embedded as inner loops inside more powerful algorithms of Wang-Landau type [16]. We leave the exploration of these newly-opened realms for future projects.

### Acknowledgments

This work was supported by an NSF CAREER award and by the US Army Research Laboratory and the US Army Research Office under contract number W911NF-12-1-0594.

## Footnotes

[1]As is well known (see e.g.[14]), for binary distributions, the Metropolis sampler that chooses a random spin and makes a proposal of flipping its value, is more efficient than the Gibbs sampler.

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
