[Supplementary Material]

# Supplemetary Material
# Auxiliary-variable Exact Hamiltonian Monte Carlo Samplers for Binary Distributions

**Ari Pakman and Liam Paninski**
Department of Statistics
Center for Theoretical Neuroscience
Grossman Center for the Statistics of Mind
Columbia University
New York, NY, 10027

## 1 Wall-crossing rate in the Gaussian augmentation

In the Gaussian augmentation, the equilibrium distribution of $(\mathbf{y}, \mathbf{q})$ in each orthant is

$$p(\mathbf{y}, \mathbf{q}|\mathbf{s}) \propto e^{-\frac{\mathbf{y} \cdot \mathbf{y}}{2}} e^{-\frac{\mathbf{q} \cdot \mathbf{q}}{2}} , \tag{1}$$

and therefore the distribution of

$$u_i = y_i^2 + q_i^2 \qquad i = 1, \ldots, d. \tag{2}$$

is $\chi_2^2$, chi-squared with two degrees of freedom. Due to conservation of energy, each $u_i$ is constant while the particle stays in an orthant and only changes if it crosses the $y_i = 0$ wall. When the particle hits the $y_i = 0$ wall, we have $u_i = q_i^2(t_i^-)$, and the particle crosses if

$$u_i > -2 \log p(-s_i, \mathbf{s}_{-i}) + 2 \log p(s_i, \mathbf{s}_{-i}) . \tag{3}$$

The probability of this event is

$$P\left[ u_i > -2 \log \left( \frac{p(-s_i, \mathbf{s}_{-i})}{p(s_i, \mathbf{s}_{-i})} \right) \right] = \begin{cases} 1 & \text{for } \frac{p(-s_i, \mathbf{s}_{-i})}{p(s_i, \mathbf{s}_{-i})} > 1 \\ 1 - C_{\chi_2^2}(-2 \log \left( \frac{p(-s_i, \mathbf{s}_{-i})}{p(s_i, \mathbf{s}_{-i})} \right)) & \text{for } \frac{p(-s_i, \mathbf{s}_{-i})}{p(s_i, \mathbf{s}_{-i})} < 1 \end{cases} \tag{4}$$

where

$$C_{\chi_2^2}(x) = 1 - e^{-\frac{x}{2}} \tag{5}$$

is the cdf of $\chi_2^2$. Inserting this expression in (4) gives

$$P\left[ u_i > -2 \log \left( \frac{p(-s_i, \mathbf{s}_{-i})}{p(s_i, \mathbf{s}_{-i})} \right) \right] = \min \left( 1, \frac{p(-s_i, \mathbf{s}_{-i})}{p(s_i, \mathbf{s}_{-i})} \right) \tag{6}$$

which is exactly the probability of acceptance in a Metropolis algorithm that samples uniformly a value for $i$ and makes a proposal of flipping the binary variable $s_i$.

## 2 Comparing the efficiency of binary samplers

We performed a more detailed comparison of the efficiency of the binary HMC sampler with Gaussian and exponential augmentations and the Metropolis sampler. As in Section 4.1, we considered a 1D Ising model with $d = 400$ and $\beta = 0.42$. The results are in Figure 1 and show that the HMC sampler with Gaussian augmentation is the most efficient of the three samplers.

Figure 1: **Efficiency comparison for binary samplers.** We considered a 1D Ising model with $d = 400$ and $\beta = 0.42$. In the Gaussian HMC sampler we considered $T = (n - 1/2)\pi$ with $n = 1, \ldots, 13$, for Metropolis we recorded the state of the chain after $d \times (n - 1/2)$ flip proposals and for the exponential HMC case we used $T's$ corresponding to similar computational costs. For each $n$ and each sampler we took 3000 samples and recorded the smallest effective sample size (ESS) among the 400 estimators $\langle s_i \rangle$. We repeated this 10 times and computed the median value of these smallest ESSs. The plot shows these values divided by the computational cost for each $n$. Note that the HMC Gaussian sampler is consistently more efficient.

## 3 Details of spike-and-slab linear regression with truncated parameters

We want to sample from the distribution

$$p(\mathbf{w}, \mathbf{y}|D, a, \tau^2) \propto e^{-\frac{1}{2}\mathbf{w}'_+\left(\mathbf{M}_+ + \tau^{-2}\right)\mathbf{w}_+ + \mathbf{r}_+ \cdot \mathbf{w}_+} e^{-\frac{\mathbf{w}_- \cdot \mathbf{w}_-}{2\tau^2}} e^{-\frac{\mathbf{y} \cdot \mathbf{y}}{2}} a^{|\mathbf{s}^+|} (1-a)^{|\mathbf{s}^-|} \tag{7}$$

where the values of $\mathbf{s}$ in the rhs are obtained from the signs of $\mathbf{y}$. Since (7) is a piecewise Gaussian distribution, we can sample from it using the methods of [1]. For this, we introduce momentum variables $q_i$ and $g_i$ associated to the coordinates $y_i$ and $w_i$ and consider the Hamiltonian

$$H = H_{\mathbf{y},\mathbf{q}} + H_{\mathbf{w},\mathbf{g}} - |\mathbf{s}^+| \log a - |\mathbf{s}^-| \log(1-a) \tag{8}$$

$$H_{\mathbf{y},\mathbf{q}} = \frac{\mathbf{y} \cdot \mathbf{y}}{2} + \frac{\mathbf{q} \cdot \mathbf{q}}{2} \tag{9}$$

$$H_{\mathbf{w},\mathbf{g}} = \frac{\mathbf{w}'_+ \mathbf{\Sigma}_+^{-1} \mathbf{w}_+}{2} - \mathbf{r}_+ \cdot \mathbf{w}_+ + \frac{\mathbf{g}'_+ \mathbf{\Sigma}_+ \mathbf{g}_+}{2} + \frac{\mathbf{w}_- \cdot \mathbf{w}_-}{2\tau^2} + \frac{\mathbf{g}_- \cdot \mathbf{g}_-}{2\tau^{-2}} \tag{10}$$

where we defined

$$\mathbf{\Sigma}_+ = \left(\mathbf{M}_+ + \tau^{-2}\right)^{-1} . \tag{11}$$

Note that we have chosen a mass matrix for $\mathbf{g}$ that depends on the orthant of $\mathbf{y}$, much like the potential terms for $\mathbf{w}$. This choice leads to decoupled equations of motion for all the coordinates, with solutions

$$y_i(t) = y_i(0)\cos(t) + q_i(0)\sin(t) , \tag{12}$$

$$w_i(t) = \mu_i + (w_i(0) - \mu_i)\cos(t) + \dot{w}_i(0)\sin(t) , \tag{13}$$

where in each orthant the components of $\boldsymbol{\mu}$ are

$$\boldsymbol{\mu}_- = 0 , \tag{14}$$

$$\boldsymbol{\mu}_+ = \mathbf{\Sigma}_+ \mathbf{r}_+ . \tag{15}$$

Each iteration of the sampling algorithm consists of sampling initial values for $\mathbf{q}$ and $\dot{\mathbf{w}}$ from

$$q_i(0) \quad \sim \quad \mathcal{N}(0,1)\,, \tag{16}$$

$$\dot{w}_i(0) \quad \sim \quad \mathcal{N}(0,\tau^2) \qquad \text{for } s_i = -1\,, \tag{17}$$

$$\dot{\mathbf{w}}_+(0) \quad \sim \quad \mathcal{N}(0,\mathbf{\Sigma}_+)\,, \tag{18}$$

and letting the particle move during a time $T$ according to the Hamiltonian (8). As before, the final coordinates belong to a Markov chain with invariant distribution $p(\mathbf{w},\mathbf{y}|D,a\tau^2)$, and are used as the initial coordinates of the next iteration. Note that it is more convenient to sample $\dot{\mathbf{w}}$ instead of $\mathbf{g}$ (related by $\dot{\mathbf{w}}_+ = \mathbf{\Sigma}_+\mathbf{g}_+$, $\dot{\mathbf{w}}_- = \tau^2\mathbf{g}_-$), because it is the former that appears in (13).

The trajectory of the particle in the $(\mathbf{y},\mathbf{w})$-space is given by (12)-(13) until some coordinate $y_j$ reaches $y_j = 0$ at time $t_j$, or, if the space of $\mathbf{w}$ is truncated, the $\mathbf{w}$ coordinates touch the boundary of their allowed space. Consider the first case and suppose that $y_j < 0$ for $t < t_j$. The conservation of energy across the $y_j = 0$ boundary implies

$$\frac{q_j^2(t_j^+)}{2} = \Delta_j + \frac{q_j^2(t_j^-)}{2}\,, \tag{19}$$

and the energy jump $\Delta_j$ depends on $\mathbf{w}$ and $\mathbf{g}$ and is given by

$$\Delta_j = -H_{\mathbf{w},\mathbf{g}}(\mathbf{s}_{-j}, s_j = +1) + H_{\mathbf{w},\mathbf{g}}(\mathbf{s}_{-j}, s_j = -1) + \log(a/(1-a))\,. \tag{20}$$

Note that the trajectory of $\mathbf{w}$, $\mathbf{g}$ is continuous at $t = t_j$, and (20) only refers to the change in the functional form of $H$ across the boundary. If (19) gives a positive value for $q_j^2(t_j^+)$, the particle crosses the $y_j = 0$ boundary, and if not, it bounces back with $q_j(t_j^+) = -q_j(t_j^-)$. In the $\mathbf{w}$-truncated case, when the $\mathbf{w}$ coordinates touch the boundary of their allowed space, the velocity $\dot{\mathbf{w}}$ is reflected off the boundary in an elastic collision, similarly to the truncated Gaussians discussed in [1].

## References

[1] Ari Pakman and Liam Paninski. Exact Hamiltonian Monte Carlo for Truncated Multivariate Gaussians. *Journal of Computational and Graphical Statistics*, 2013, arXiv:1208.4118.