[Reviews · NeurIPS 2013]

Submitted by Assigned_Reviewer_1

The authors propose a new exact Hamiltonian Monte Carlo (HMC) samplers to sample from binary distributions. Their method uses the fact that for a suitable auxiliary variable construction, it is possible to solve exactly the equations of motion appearing in HMC. This is neat. This approach has been pioneered to the best of my knowledge in a recent paper of Paninski and it's a natural follow-up.

To the best of my knowledge, this is original and interesting. However the paper is unfortunately not as clear as it could/should be and the writing has clearly been rushed: there are embarrassing typos (e.g. p(y)=p(y,s)in equations 4 and 5, supplemetary material).

The paper organization should be improved. We have a long section 3 dedicated to spike and slab regression with truncated parameters (interesting but having simulation for the standard case would have been interesting too) and simulation results appear in section 4.3 while section 4.1 and 4.2 are dedicated to the Ising model... It would have been better to expand section 2 and rewrite it very clearly with the method described in a pseudo-algorithm form and merge sections 3 and 4 together (e.g. do we need for example equations (26) and (27))

The simulation section is a bit deficient. It would have been also interesting to study at least experimentally the sensitivity of the algorithm to the travel time T. It's not clear why for various applications how the authors have chosen the values of T. Giving some recommendations to the potential users would have been useful.
Summary: There is a neat idea in this paper and I think the method proposed here is potentially useful. I only gave a 7 to the paper because it is unfortunately not very well-written.

Submitted by Assigned_Reviewer_3

The authors present a new method for sampling from binary distributions using an exact Hamiltonian Monte Carlo scheme and demonstrate its applicability on linear and probit regression models with spike and slab priors and truncated parameters, problems that consist of mixtures of continuous and binary variables.

The paper is polished, well written and generally very clear. This work provides a solution that may be applied to general binary or mixed binary/continuous distributions, building upon previous work using a different approach which was only applicable to Markov random fields. As far as I am aware this is a novel piece of work and I very much enjoyed reading it. As such I only have a few fairly minor comments to make:

Section 3.1: Slightly confusing for the reader that D is used for the data, but then without warning z is immediately used to represent the data.

Examples section 4: A variety of examples are shown with nice graphs showing the sampler output. I think the reader would appreciate a more rigorous quantitative comparison in terms of the effective sample size, perhaps also normalised by the computational time required to draw the samples. Of course this must be done carefully trying to minimise the effect of coding differences in implementation, but I think it would strengthen the argument for the adoption of such samplers.

Page 4, line 209: Typo, "Horsehoe" -> "Horseshoe"
Summary: An interesting paper, well presented and of reasonable interest to the wider community.

Submitted by Assigned_Reviewer_6

This paper describes a method for relaxing discrete sampling problems into continuous ones that are suitable for Hamiltonian Monte Carlo. The idea is to divide the continuous space into orthants, creating a one-to-one correspondence between the possible values of the discrete space and orthants in the continuous space. The complication that arises is that the relaxation has noncontinuous jumps at the boundaries between orthants, which the authors address by augmenting Hamiltonian dynamics to allow the sample to either reflect off or jump on top of cliffs, depending on its energy. A nice effect of this construction is within the orthant, the problem is a constrained quadratic, for which recent work has pointed out that the dynamics can be computed exactly, eliminating the need for numerical integration.

This is a creative and nice application of recent work on exact Hamiltonian samplers for log-quadratic densities. Spike-and-slab models are a great application of this technique, and an important model, and the results show that this method has promise as an effective sampling algorithm for this model.

I have a question about the validity of the method. It's clear to me that with energy barriers added, the dynamics is reversible and preserves the value of the Hamiltonian. I cannot immediately see why the dynamics conserves volume. Obviously it does so away from the boundaries, but it is less clear to me why the dynamics conserves volume at the energy barriers. Could the authors please expand on this?

Assuming that the method is valid, it's an interesting paper that merits publication.

Minor comments:

* lines 108ff: A possible comment that might help the reader's intuition: The equation (9) has a solution iff the right hand side is positive. So if Delta is positive, meaning that the new discrete state would have higher probability than the current one, then the coordinate always crosses the boundary, i.e., we always move to the new discrete state. In the physical analogy of HMC, the dynamics reaches the top of a cliff, so it always falls downward. On the other hand, if Delta is negative, then the new discrete state has lower probability, so we only switch to it if the sampler has enough energy, i.e., if q^2_j (t_j^-)/2 > -Delta.

==========
AFTER REBUTTAL

1. Thanks for your explanation about the boundary effects.

2. Perhaps a bit self-indulgent, re-reading my review, I thought of a perhaps better way to express my intuition at the end.

"On the other hand, if Delta is negative, then the new discrete state has lower probability. Because the sampler has to be reversible, we're only allowed to roll up the cliff if there exists a reverse trajectory that could have been generated by the dynamics. In the reverse trajectory, the particle gains energy when it falls down the cliff, so this creates a lower bound on the energy of particles that jump up the cliff."

Feel free to not use this if you feel it doesn't help.
Summary: Assuming that the method is valid, it's an interesting paper that merits publication.
Author Feedback

Author rebuttal: We would like to thank the reviewers for their comments and suggestions. Below we address their comments in detail:

From Assigned_Reviewer_1

> there are embarrassing typos (e.g. p(y)=p(y,s)in equations 4 and 5, supplementary material).

The identity "p(y)=p(y,s)" is not a typo, but follows from the definition of p(y). Note that the rhs only depends on y, since s=sign(y). To make this clear, perhaps the sum in (4) should be over s' instead of over s. We will change this in a revised version.


> We have a long section 3 dedicated to spike and slab regression with truncated parameters (interesting but having simulation for the standard case would have been interesting too)

As mentioned in line 247ff, in the standard case (without truncated parameters) one can integrate out the weights and just sample the binary variables. This gives a Rao-Blackwellized result, with lower variance. As mentioned there, our approach is most useful precisely when such an exact integration is not possible because the parameter region is truncated.


> and simulation results appear in section 4.3 while section 4.1 and 4.2 are dedicated to the Ising model...
> It would have been better to expand section 2 and rewrite it very clearly with the method described in a pseudo-algorithm form and merge sections 3 and 4 together
> (e.g. do we need for example equations (26) and (27))

A problem with merging sections 3 and 4 is that the latter contains Ising model examples that would belong in section 2, and adding examples already in Section 2 would break the continuity of the theoretical exposition. As for the equations, since equation (28) is quite central to the posterior analysis, we thought a reader would find useful to see its derivation from (26) and (27).

Summarizing the method in a pseudo-algorithm form is a good idea, we will add it in a revised version.


> The simulation section is a bit deficient. It would have been also interesting to study at least experimentally the sensitivity of the algorithm to the travel time T. It's not clear why for various applications how the authors have chosen the values of T. Giving some recommendations to the potential users would have been useful.

Thanks, great suggestion. As usual with the HMC method, the value of T giving the best results requires experimenting in each particular problem. We will address this point in a revised version.




From Assigned_Reviewer_3

> Section 3.1: Slightly confusing for the reader that D is used for the data, but then without warning z is immediately used to represent the data.

Thanks. We will correct the notation in a revised version.

> Examples section 4: I think the reader would appreciate a more rigorous quantitative comparison in terms of the effective sample size, perhaps also normalised by the computational time required to draw the samples.

That would be a good addition, we will add such a comparison in a revised version.

> Page 4, line 209: Typo, "Horsehoe" -> "Horseshoe"

Thanks, will correct that.



From Assigned_Reviewer_6

> it is less clear to me why the dynamics conserves volume at the energy barriers. Could the authors please expand on this?

Thanks, this is an important point. A region of phase space with spatial coordinates near an orthant boundary splits into two regions, according to whether the coordinate is reflected or crosses the boundary. One region will have the momentum reflected and the other transforms discretely, its volume conservation following from the limit of a smooth but steep potential jump. Both transformations preserve volume and hence the total volume is conserved. This argument applies both to the binary and mixed continuous-binary cases. We will be happy to elaborate on these points in a revised version.


> lines 108ff: A possible comment that might help the reader's intuition: The equation (9) has a solution iff the right hand side is positive. So if Delta is positive, meaning that the new discrete state would have higher probability
> than the current one, then the coordinate always crosses the boundary, i.e., we always move to the new discrete state. In the physical analogy of HMC, the dynamics reaches the top of a cliff, so it always falls downward.
> On the other hand, if Delta is negative, then the new discrete state has lower probability, so we only switch to it if the sampler has enough energy, i.e., if q^2_j (t_j^-)/2 > -Delta.

Thanks, we will add a comment along these lines in a revised version.